# Sanity Checks for Saliency Maps

**Julius Adebayo,**[*] **Justin Gilmer**[♯]**, Michael Muelly**[♯]**, Ian Goodfellow**[♯]**, Moritz Hardt**[♯†]**, Been Kim**[♯]
juliusad@mit.edu, {gilmer,muelly,goodfellow,mrtz,beenkim}@google.com
[♯]Google Brain
[†]University of California Berkeley

## Abstract

Saliency methods have emerged as a popular tool to highlight features in an input deemed relevant for the prediction of a learned model. Several saliency methods have been proposed, often guided by visual appeal on image data. In this work, we propose an actionable methodology to evaluate what kinds of explanations a given method can and cannot provide. We find that reliance, solely, on visual assessment can be misleading. Through extensive experiments we show that some existing saliency methods are independent both of the model and of the data generating process. Consequently, methods that fail the proposed tests are inadequate for tasks that are sensitive to either data or model, such as, finding outliers in the data, explaining the relationship between inputs and outputs that the model learned, and debugging the model. We interpret our findings through an analogy with edge detection in images, a technique that requires neither training data nor model. Theory in the case of a linear model and a single-layer convolutional neural network supports our experimental findings[2].

## 1 Introduction

As machine learning grows in complexity and impact, much hope rests on explanation methods as tools to elucidate important aspects of learned models [1, 2]. Explanations could potentially help satisfy regulatory requirements [3], help practitioners debug their model [4, 5], and perhaps, reveal bias or other unintended effects learned by a model [6, 7]. *Saliency methods*[3] are an increasingly popular class of tools designed to highlight relevant features in an input, typically, an image. Despite much excitement, and significant recent contribution [8–21], the valuable effort of explaining machine learning models faces a methodological challenge: *the difficulty of assessing the scope and quality of model explanations.* A paucity of principled guidelines confound the practitioner when deciding between an abundance of competing methods.

We propose an actionable methodology based on randomization tests to evaluate the adequacy of explanation approaches. We instantiate our analysis on several saliency methods for image classification with neural networks; however, our methodology applies in generality to any explanation approach. Critically, our proposed randomization tests are easy to implement, and can help assess the suitability of an explanation method for a given task at hand.

In a broad experimental sweep, we apply our methodology to numerous existing saliency methods, model architectures, and data sets. To our surprise, *some widely deployed saliency methods are independent of both the data the model was trained on, and the model parameters.* Consequently,

---

[*]Work done during the Google AI Residency Program.
[2]All code to replicate our findings will be available here: https://goo.gl/hBmhDt
[3]We refer here to the broad category of visualization and attribution methods aimed at interpreting trained models. These methods are often used for interpreting deep neural networks particularly on image data.

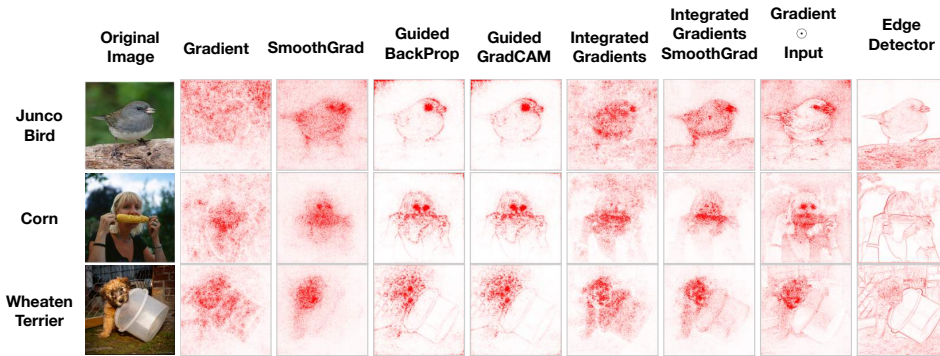

Figure 1: **Saliency maps for some common methods compared to an edge detector.** Saliency masks for 3 inputs for an Inception v3 model trained on ImageNet. We see that an edge detector produces outputs that are strikingly similar to the outputs of some saliency methods. In fact, edge detectors can also produce masks that highlight features which coincide with what appears to be relevant to a model's class prediction. We find that the methods most similar (see Appendix for SSIM metric) to an edge detector, i.e., Guided Backprop and its variants, show minimal sensitivity to our randomization tests.

these methods are incapable of assisting with tasks that depend on the model, such as debugging the model, or tasks that depend on the relationships between inputs and outputs present in the data.

To illustrate the point, Figure 1 compares the output of standard saliency methods with those of an edge detector. The edge detector does not depend on model or training data, and yet produces results that bear visual similarity with saliency maps. This goes to show that visual inspection is a poor guide in judging whether an explanation is sensitive to the underlying model and data.

Our methodology derives from the idea of a statistical randomization test, comparing the natural experiment with an artificially randomized experiment. We focus on two instantiations of our general framework: a *model parameter randomization test*, and a *data randomization test*.

**The model parameter randomization test** compares the output of a saliency method on a trained model with the output of the saliency method on a randomly initialized untrained network of the same architecture. If the saliency method depends on the learned parameters of the model, we should expect its output to differ substantially between the two cases. Should the outputs be similar, however, we can infer that the saliency map is insensitive to properties of the model, in this case, the model parameters. In particular, the output of the saliency map would not be helpful for tasks such as *model debugging* that inevitably depend on the model parameters.

**The data randomization test** compares a given saliency method applied to a model trained on a labeled data set with the method applied to the same model architecture but trained on a copy of the data set in which we randomly permuted all labels. If a saliency method depends on the labeling of the data, we should again expect its outputs to differ significantly in the two cases. An insensitivity to the permuted labels, however, reveals that the method does not depend on the relationship between instances (e.g. images) and labels that exists in the original data.

Speaking more broadly, any explanation method admits a set of *invariances*, i.e., transformations of data and model that do not change the output of the method. If we discover an invariance that is incompatible with the requirements of the task at hand, we can safely reject the method. As such, our tests can be thought of as *sanity checks* to perform before deploying a method in practice.

**Our contributions**

1. We propose two concrete, easy to implement tests for assessing the scope and quality of explanation methods: the model parameter randomization test, and the data randomization test. These tests apply broadly to explanation methods.

2. We conduct extensive experiments with several explanation methods across data sets and model architectures, and find, consistently, that some of the methods tested are independent of both the model parameters and the labeling of the data that the model was trained on.

3. Of the methods we tested, Gradients & GradCAM pass the sanity checks, while Guided BackProp & Guided GradCAM fail. In the other cases, we observe a visual perception versus ranking dichotomy, which we describe in our results.

4. Consequently, our findings imply that the saliency methods that fail our tests are incapable of supporting tasks that require explanations that are faithful to the model or the data generating process.

5. We interpret our findings through a series of analyses of linear models and a simple 1-layer convolutional sum-pooling architecture, as well as a comparison with edge detectors.

## 2 Methods and Related Work

In our formal setup, an *input* is a vector $x \in \mathbb{R}^d$. A *model* describes a function $S \colon \mathbb{R}^d \to \mathbb{R}^C$, where $C$ is the number of *classes* in the classification problem. An explanation method provides an *explanation map* $E \colon \mathbb{R}^d \to \mathbb{R}^d$ that maps inputs to objects of the same shape.

We now briefly describe some of the explanation methods we examine. The supplementary materials contain an in-depth overview of these methods. Our goal is not to exhaustively evaluate all prior explanation methods, but rather to highlight how our methods apply to several cases of interest.

The **gradient explanation** for an input $x$ is $E_{\mathrm{grad}}(x) = \frac{\partial S}{\partial x}$ [8, 22, 23]. The gradient quantifies how much a change in each input dimension would a change the predictions $S(x)$ in a small neighborhood around the input.

**Gradient $\odot$ Input.** Another form of explanation is the element-wise product of the input and the gradient, denoted $x \odot \frac{\partial S}{\partial x}$, which can address "gradient saturation", and reduce visual diffusion [13].

**Integrated Gradients (IG)** also addresses gradient saturation by summing over scaled versions of the input [14]. IG for an input $x$ is defined as $E_{\mathrm{IG}}(x) = (x - \bar{x}) \times \int_0^1 \frac{\partial S(\bar{x} + \alpha(x - \bar{x}))}{\partial x} d\alpha$, where $\bar{x}$ is a "baseline input" that represents the absence of a feature in the original input $x$.

**Guided Backpropagation (GBP)** [9] builds on the "DeConvNet" explanation method [10] and corresponds to the gradient explanation where negative gradient entries are set to zero while back-propagating through a ReLU unit.

**Guided GradCAM.** Introduced by Selvaraju et al. [19], GradCAM explanations correspond to the gradient of the class score (logit) with respect to the feature map of the last convolutional unit of a DNN. For pixel level granularity GradCAM can be combined with Guided Backpropagation through an element-wise product.

**SmoothGrad (SG)** [16] seeks to alleviate noise and visual diffusion [14, 13] for saliency maps by averaging over explanations of noisy copies of an input. For a given explanation map $E$, SmoothGrad is defined as $E_{\mathrm{sg}}(x) = \frac{1}{N} \sum_{i=1}^N E(x + g_i)$, where noise vectors $g_i \sim \mathcal{N}(0, \sigma^2))$ are drawn i.i.d. from a normal distribution.

### 2.1 Related Work

**Other Methods & Similarities.** Aside gradient-based approaches, other methods 'learn' an explanation per sample for a model [20, 17, 12, 15, 11, 21]. More recently, M. Ancona [24] showed that for ReLU networks (with zero baseline and no biases) the $\epsilon$-LRP and DeepLift (Rescale) explanation methods are equivalent to the $input \odot gradient$. Similarly, Lundberg and Lee [18] proposed SHAP explanations which approximate the shapley value and unify several existing methods.

**Fragility.** Ghorbani et al. [25] and Kindermans et al. [26] both present attacks against saliency methods; showing that it is possible to manipulate derived explanations in unintended ways. Nie et al. [27] theoretically assessed backpropagation based methods and found that Guided BackProp and DeconvNet, under certain conditions, are invariant to network reparamaterizations, particularly random Gaussian initialization. Specifically, they show that Guided BackProp and DeconvNet both seem to be performing partial input recovery. Our findings are similar for Guided BackProp and its variants. Further, our work differs in that we propose actionable sanity checks for assessing explanation approaches. Along similar lines, Mahendran and Vedaldi [28] also showed that some backpropagation-based saliency methods lack neuron discriminativity.

**Current assessment methods.** Both Samek et al. [29] and Montavon et al. [30] proposed an input perturbation procedure for assessing the quality of saliency methods. Dabkowski and Gal [17] proposed an entropy-based metric to quantify the amount of relevant information an explanation mask captures. Performance of a saliency map on an object localization task has also been used for assessing saliency methods. Montavon et al. [30] discuss explanation continuity and selectivity as measures of assessment.

**Randomization.** Our label randomization test was inspired by the work of Zhang et al. [31], although we use the test for an entirely different purpose.

## 2.2 Visualization & Similarity Metrics

We discuss our visualization approach and overview the set of metrics used in assessing similarity between two explanations.

**Visualization.** We visualize saliency maps in two ways. In the first case, *absolute-value* (ABS), we take absolute values of a normalized[4] map. For the second case, *diverging* visualization, we leave the map as is, and use different colors to show positive and negative importance.

**Similarity Metrics.** For quantitative comparison, we rely on the following metrics: Spearman rank correlation with absolute value (absolute value), Spearman rank correlation without absolute value (diverging), the structural similarity index (SSIM), and the Pearson correlation of the histogram of gradients (HOGs) derived from two maps. We compute the SSIM and HOGs similarity metric on ImageNet examples without absolute values.[5] These metrics capture a broad notion of similarity; however, quantifying human visual perception is still an active area of research.

## 3 Model Parameter Randomization Test

The parameter settings of a model encode what the model has learned from the data during training, and determine test set performance. Consequently, for a saliency method to be useful for debugging a model, it ought to be sensitive to model parameters.

As an illustrative example, consider a linear function of the form $f(x) = w_1 x_1 + w_2 x_2$ with input $x \in \mathbb{R}^2$. A gradient-based explanation for the model's behavior for input $x$ is given by the parameter values $(w_1, w_2)$, which correspond to the sensitivity of the function to each of the coordinates. Changes in the model parameters therefore change the explanation.

Our proposed model parameter randomization test assesses an explanation method's sensitivity to model parameters. We conduct two kinds of randomization. First we randomly re-initialize all weights of the model both completely and in a cascading fashion. Second, we independently randomize a single layer at a time while keeping all others fixed. In both cases, we compare the resulting explanation from a network with random weights to the one obtained with the model's original weights.

### 3.1 Cascading Randomization

**Overview.** In the cascading randomization, we randomize the weights of a model starting from the top layer, successively, all the way to the bottom layer. This procedure destroys the learned weights from the top layers to the bottom ones. Figure 2 visualizes the cascading randomization for several saliency methods. In Figures 3 and 4, we show the Spearman metrics as well as the SSIM and HOGs similarity metrics.

**The gradient shows sensitivity while Guided BackProp is invariant.** We find that the gradient map is sensitive to model parameters. We also observe sensitivity for the GradCAM masks. On the other hand, across all architectures and datasets, Guided BackProp and Guided GradCAM show no change regardless of model degradation.

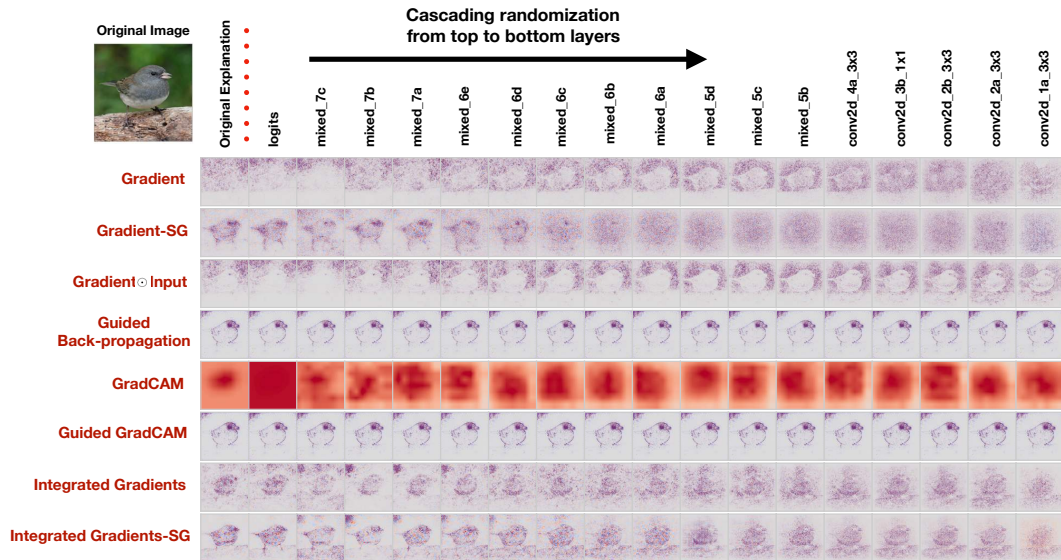

Figure 2: **Cascading randomization on Inception v3 (ImageNet).** Figure shows the original explanations (first column) for the Junco bird. Progression from left to right indicates complete randomization of network weights (and other trainable variables) up to that 'block' inclusive. We show images for 17 blocks of randomization. Coordinate (Gradient, mixed_7b) shows the gradient explanation for the network in which the top layers starting from Logits up to mixed_7b have been reinitialized. The last column corresponds to a network with completely reinitialized weights.

**The danger of the visual assessment.** On visual inspection, we find that integrated gradients and gradient⊙input show a remarkable visual similarity to the original mask. In fact, from Figure 2, it is still possible to make out the structure of the bird even after multiple blocks of randomization. This visual similarity is reflected in the rank correlation with absolute value (Figure 3-Top), SSIM, and the HOGs metric (Figure 4). However, re-initialization disrupts the sign of the map, so that the Spearman rank correlation without absolute values goes to zero (Figure 3-Bottom) almost as soon as the top layers are randomized. This observed visual perception versus numerical ranking dichotomy indicates that naive visual inspection of the masks does not distinguish networks of similar structure but widely differing parameters. We explain the source of this phenomenon in our discussion section.

## 3.2 Independent Randomization

**Overview.** As a different form of the model parameter randomization test, we conduct an independent layer-by-layer randomization with the goal of isolating the dependence of the explanations by layer. Consequently, we can assess the dependence of saliency masks on lower versus higher layer weights.

**Results.** We observe a correspondence between the results from the cascading and independent layer randomization experiments (see Figures **??**, **??**, **??**, and **??** in the Appendix). As previously observed, Guided Backprop and Guided GradCAM masks remain almost unchanged regardless of the layer that is independently randomized across all networks. Similarly, we observe that the structure of the input is maintained, visually, for the gradient⊙input and Integrated Gradient methods.

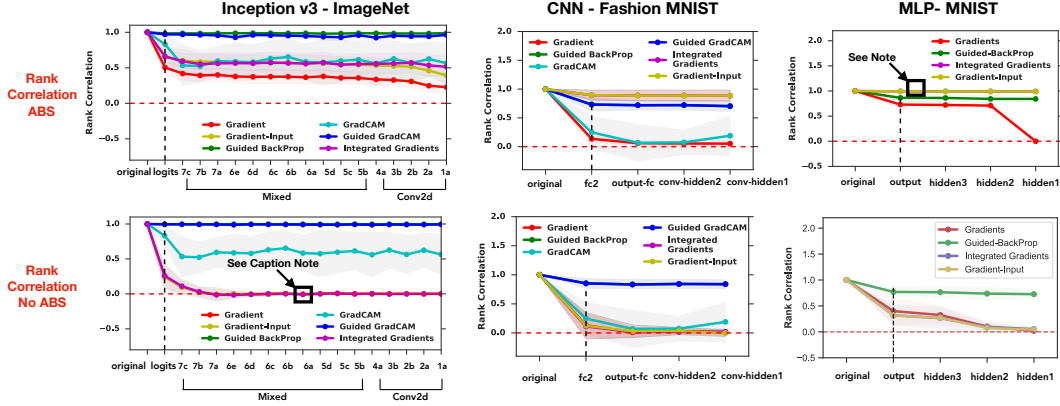

Figure 3: **Similarity Metrics for Cascading Randomization.** We show results for Inception v3 on ImageNet, CNN on Fashion MNIST, and MLP on MNIST. See appendix for MLP on Fashion MNIST and CNN on MNIST. In all plots, y axis is the rank correlation between original explanation and the randomized explanation derived for randomization up to that layer/block, while the x axis corresponds to the layers/blocks of the DNN starting from the output layer. The vertical black dashed line indicates where successive randomization of the network begins, which is at the top layer. **Top**: Spearman Rank correlation with absolute values, **Bottom**: Spearman Rank correlation without absolute values. **Caption Note**: For Inception v3 on ImageNet no ABS, the IG, gradient-input, and gradients all coincide. For MLP-MNIST IG and gradient-input coincide.

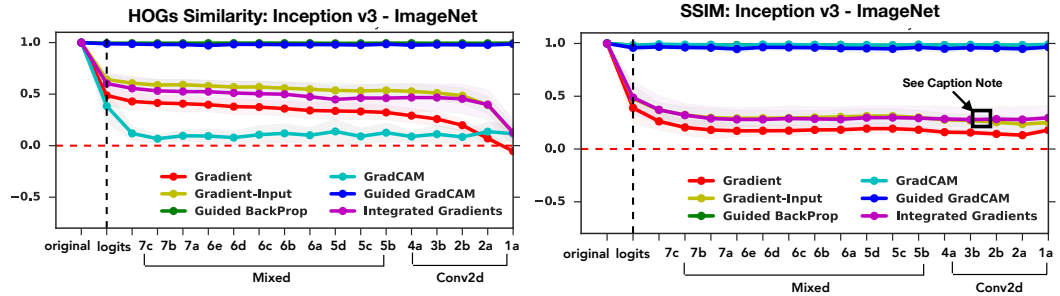

Figure 4: **Similarity Metrics for Cascading Randomization.** Figure showing HOGs similarity and SSIM between original input masks and the masks generated as the Inception v3 is randomized in a cascading manner. **Caption Note**: For SSIM: Inception v3 - ImageNet, IG and gradient⊙input coincide, while GradCAM, Guided GradCAM, and Guided BackProp are clustered together at the top.

# 4 Data Randomization Test

The feasibility of accurate prediction hinges on the relationship between instances (e.g., images) and labels encoded by the data. If we artificially break this relationship by randomizing the labels, no predictive model can do better than random guessing. Our data randomization test evaluates the sensitivity of an explanation method to the relationship between instances and labels. An explanation method insensitive to randomizing labels cannot possibly explain mechanisms that depend on the relationship between instances and labels present in the data generating process. For example, if an explanation did not change after we randomly assigned diagnoses to CT scans, then evidently it did *not* explain anything about the relationship between a CT scan and the correct diagnosis in the first place (see [32] for an application of Guided BackProp as part of a pipeline for shadow detection in 2D Ultrasound).

In our data randomization test, we permute the training labels and train a model on the *randomized* training data. A model achieving high training accuracy on the randomized training data is forced to *memorize* the randomized labels without being able to exploit the original structure in the data. As it

turns out, state-of-the art deep neural networks can easily fit random labels as was shown in Zhang et al. [31].

In our experiments, we permute the training labels for each model and data set pair, and train the model to greater than $95\%$ training set accuracy. Note that the test accuracy is never better than randomly guessing a label (up to sampling error). For each resulting model, we then compute explanations on the same test bed of inputs for a model trained with true labels and the corresponding model trained on randomly permuted labels.

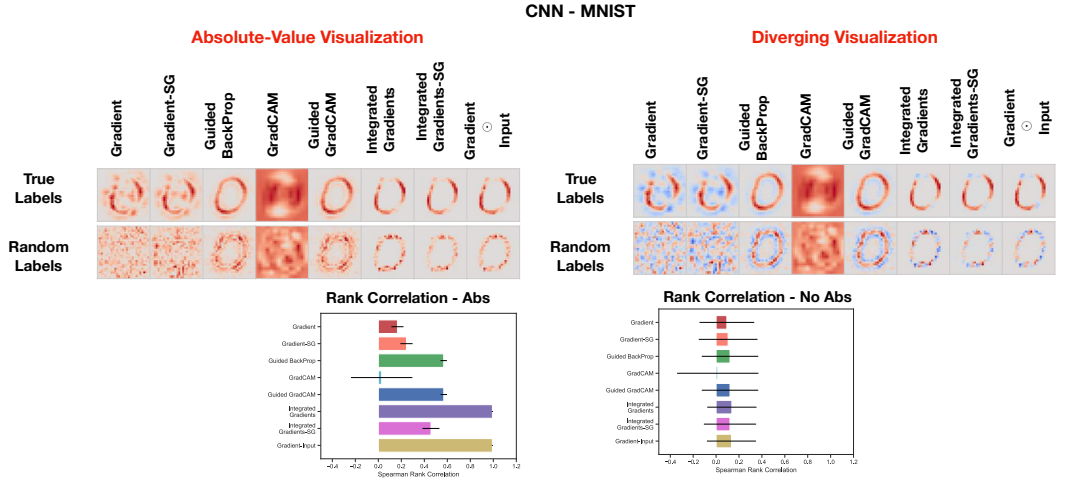

Figure 5: **Explanation for a true model vs. model trained on random labels. Top Left**: Absolute-value visualization of masks for digit 0 from the MNIST test set for a CNN. **Top Right**: Saliency masks for digit 0 from the MNIST test set for a CNN shown in diverging color. **Bottom Left**: Spearman rank correlation (with absolute values) bar graph for saliency methods. We compare the similarity of explanations derived from a model trained on random labels, and one trained on real labels. **Bottom Right**: Spearman rank correlation (without absolute values) bar graph for saliency methods for MLP. See appendix for corresponding figures for CNN, and MLP on Fashion MNIST.

**Gradient is sensitive.** We find, again, that gradients, and its smoothgrad variant, undergo substantial changes. In addition, the GradCAM masks also change becoming more disconnected.

**Sole reliance on visual inspection can be misleading.** For Guided BackProp, we observe a visual change; however, we find that the masks still highlight portions of the input that would seem plausible, given correspondence with the input, on naive visual inspection. For example, from the diverging masks (Figure 5-Right), we see that the Guided BackProp mask still assigns positive relevance across most of the digit for the network trained on random labels.

For gradient⊙input and integrated gradients, we also observe visual changes in the masks obtained, particularly, in the sign of the attributions. Despite this, the input structure is still clearly prevalent in the masks. The effect observed is particularly prominent for sparse inputs like MNIST where the background is zero; however, we observe similar effects for Fashion MNIST (see Appendix). With visual inspection alone, it is not inconceivable that an analyst could confuse the integrated gradient and gradient⊙input masks derived from a network trained on random labels as legitimate.

## 5  Discussion

We now take a step back to interpret our findings. First, we discuss the influence of the model architecture on explanations derived from NNs. Second, we consider methods that approximate an element-wise product of input and gradient, as several local explanations do [33, 18]. We show, empirically, that the input "structure" dominates the gradient, especially for sparse inputs. Third, we explain the observed behavior of the gradient explanation with an appeal to linear models. We then consider a single 1-layer convolution with sum-pooling architecture, and show that saliency explanations for this model mostly capture edges. Finally, we return to the edge detector and make comparisons between the methods that fail our sanity checks and an edge detector.

## 5.1 The role of model architecture as a prior

The architecture of a deep neural network has an important effect on the representation derived from the network. A number of results speak to the strength of randomly initialized models as classification priors [34, 35]. Moreover, randomly initialized networks trained on a single input can perform tasks like denoising, super-resolution, and in-painting [36] without additional training data. These prior works speak to the fact that randomly initialized networks correspond to non-trivial representations. Explanations that do not depend on model parameters or training data might still depend on the model architecture and thus provide some useful information about the prior incorporated in the model architecture. However, in this case, the explanation method should only be used for tasks where we believe that knowledge of the model architecture on its own is sufficient for giving useful explanations.

## 5.2 Element-wise input-gradient products

A number of methods, e.g., $\epsilon$-LRP, DeepLift, and integrated gradients, approximate the element-wise product of the input and the gradient (on a piecewise linear function like ReLU). To gain further insight into our findings, we can look at what happens to the input-gradient product $E(x) = x \odot \frac{\partial S}{\partial x}$, if the input is kept fixed, but the gradient is randomized. To do so, we conduct the following experiment. For an input $x$, sample two random vectors $u, v$ (we consider both the truncated normal and uniform distributions) and consider the element-wise product of $x$ with $u$ and $v$, respectively, i.e., $x \odot u$, and $x \odot v$. We then look at the similarity, for all the metrics considered, between $x \odot u$ and $x \odot v$ as noise increases. We conduct this experiment on ImageNet samples. We observe that the input does indeed dominate the product (see Figure **??** in Appendix). We also observe that the input dominance persists even as the noisy gradient vectors change drastically. This experiment indicates that methods that approximate the "input-times-gradient" could conceivably mostly return the input, in cases where the gradients look visually noisy as they tend to do.

## 5.3 Analysis for simple models

To better understand our findings, we analyze the output of the saliency methods tested on two simple models: a linear model and a 1-layer sum pooling convolutional network. We find that the output of the saliency methods, on a linear model, returns a coefficient that intuitively measures the sensitivity of the model with respect to that variable. However, these methods applied to a random convolution seem to result in visual artifacts that are akin to an edge detector.

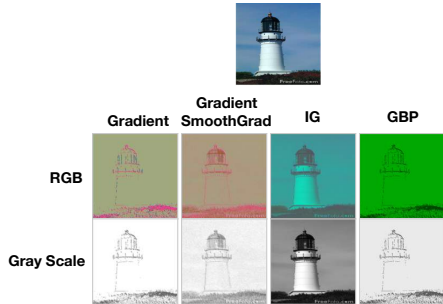

Figure 6: Explanations derived for the 1-layer Sum-Pooling Convolution architecture. We show gradient, SmoothGrad, Integrated Gradients, and Guided Back-Prop explanations. (See Appendix for Similarity Metrics).

**Linear Model.** Consider a linear model $f : \mathbb{R}^d \to \mathbb{R}$ defined as $f(x) = w \cdot x$ where $w \in \mathbb{R}^d$ are the model weights. For gradients we have $E_{\mathrm{grad}}(x) = \frac{\partial(w \cdot x)}{\partial x} = w$. Similarly for SmoothGrad we have $E_{\mathrm{sg}}(x) = w$ (the gradient is independent of the input, so averaging gradients over noisy inputs yields the same model weight). Integrated Gradients reduces to "gradient $\odot$ input" for this case:

$$E_{IG}(x) = (x - \bar{x}) \odot \int_0^1 \frac{\partial f(\bar{x} + \alpha(x - \bar{x}))}{\partial x} d\alpha \quad = (x - \bar{x}) \odot \int_0^1 w\alpha d\alpha = (x - \bar{x}) \odot w/2 \,.$$

Consequently, we see that the application of the basic gradient method to a linear model will pass our sanity check. Gradients on a random model will return an image of white noise, while integrated gradients will return a noisy version of the input image. We did not consider Guided Backprop and GradCAM here because both methods are not defined for the linear model considered above.

**1 Layer Sum-Pool Conv Model.** We now show that the application of these same methods to a 1-layer convolutional network may result in visual artifacts that can be misleading unless further

analysis is done. Consider a single-layer convolutional network applied to a grey-scale image $x \in \mathbb{R}^{n \times n}$. Let $w \in \mathbb{R}^{3 \times 3}$ denote the $3 \times 3$ convolutional filter, indexed as $w_{ij}$ for $i, j \in \{-1, 0, 1\}$. We denote by $w * x \in \mathbb{R}^{n \times n}$ the output of the convolution operation on the image $x$. Then the output of this network can be written as $l(x) = \sum_{i=1}^{n} \sum_{j=1}^{n} \sigma(w * x)_{ij}$, where $\sigma$ is the ReLU non-linearity applied point-wise. In particular, this network applies a single 3x3 convolutional filter to the input image, then applies a ReLU non-linearity and finally sum-pools over the entire convolutional layer for the output. This is a similar architecture to the one considered in [34]. As shown in Figure 6, we see that different saliency methods do act like edge detectors. This suggests that the convolutional structure of the network is responsible for the edge detecting behavior of some of these saliency methods.

To understand why saliency methods applied to this simple architecture visually appear to be edge detectors, we consider the closed form of the gradient $\frac{\partial}{\partial x_{ij}} l(x)$. Let $a_{ij} = \mathbf{1}\{(w * x)_{ij} \geq 0\}$ indicate the activation pattern of the ReLU units in the convolutional layer. Then for $i, j \in [2, n-1]$ we have

$$\frac{\partial}{\partial x_{ij}} l(x) = \sum_{k=-1}^{1} \sum_{l=-1}^{1} \sigma'((w * x)_{i+k,j+l}) w_{kl} = \sum_{k=-1}^{1} \sum_{l=-1}^{1} a_{i+k,j+l} w_{kl}$$

(Recall that $\sigma'(x) = 0$ if $x < 0$ and $1$ otherwise). This implies that the $3 \times 3$ activation pattern local to pixel $x_{ij}$ uniquely determines $\frac{\partial}{\partial x_{ij}}$. It is now clear why edges will be visible in the produced saliency mask — regions in the image corresponding to an "edge" will have a distinct activation pattern from surrounding pixels. In contrast, pixel regions of the image which are more uniform will all have the same activation pattern, and thus the same value of $\frac{\partial}{\partial x_{ij}} l(x)$. Perhaps a similar principle applies for stacked convolutional layers.

### 5.4 The case of edge detectors.

An *edge detector*, roughly speaking, is a classical tool to highlight sharp transitions in an image. Notably, edge detectors are typically untrained and do not depend on any predictive model. They are solely a function of the given input image. As some of the saliency methods we saw, edge detection is invariant under model and data transformations.

In Figure 1 we saw that edge detectors produce images that are strikingly similar to the outputs of some saliency methods. In fact, edge detectors can also produce pictures that highlight features which coincide with what appears to be relevant to a model's class prediction. However, here the human observer is at risk of confirmation bias when interpreting the highlighted edges as an explanation of the class prediction. In Figure **??** (In Appendix), we show a qualitative comparison of saliency maps of an input image with the same input image multiplied element-wise by the output of an edge detector. The result indeed looks strikingly similar, illustrating that saliency methods mostly use the edges of the image.

While edge detection is a fundamental and useful image processing technique, it is typically not thought of as an explanation method, simply because it involves no model or training data. In light of our findings, it is not unreasonable to interpret some saliency methods as implicitly implementing unsupervised image processing techniques, akin to edge detection, segmentation, or denoising. To differentiate such methods from model-sensitive explanations, visual inspection is insufficient.

## 6 Conclusion and future work

The goal of our experimental method is to give researchers guidance in assessing the scope of model explanation methods. We envision these methods to serve as sanity checks in the design of new model explanations. Our results show that visual inspection of explanations alone can favor methods that may provide compelling pictures, but lack sensitivity to the model and the data generating process.

Invariances in explanation methods give a concrete way to rule out the adequacy of the method for certain tasks. We primarily focused on invariance under model randomization, and label randomization. Many other transformations are worth investigating and can shed light on various methods we did and did not evaluate. Along these lines, we hope that our paper is a stepping stone towards a more rigorous evaluation of new explanation methods, rather than a verdict on existing methods.

## Acknowledgments

We thank the Google PAIR team for open source implementation of the methods used in this work. We thank Martin Wattenberg and other members of the Google Brain team for critical feedback that helped improved the work. Lastly, we thank anonymous reviewers for feedback that helped improve the manuscript.

## Footnotes

[4]We normalize the maps to the range $[-1.0, 1.0]$. Normalizing in this manner potentially ignores peculiar characteristics of some saliency methods. For example, Integrated gradients has the property that the attributions sum up to the output value. This property cannot usually be visualized. We contend that such properties will not affect the manner in which the output visualizations are perceived.

[5]See appendix for a discussion on calibration of these metrics.

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
