[Supplementary Material]

# Appendix

## A   Explanation Methods

We now provide additional overview of the different saliency methods that we assess in this work. As described in the main text, an *input* is a vector $x \in \mathbb{R}^d$. A *model* describes a function $S \colon \mathbb{R}^d \to \mathbb{R}^C$, where $C$ is the number of *classes* in the classification problem. An explanation method provides an *explanation map* $E \colon \mathbb{R}^d \to \mathbb{R}^d$ that maps inputs to objects of the same shape. Each dimension then correspond to the 'relevance' or 'importance' of that dimension to the final output, which is often a class-specific score as specified above.

### A.1   Gradient with respect to input

This corresponds to the gradient of the scalar logit for a particular class wrt to the input.

$$E_{\text{grad}}(x) = \frac{\partial S}{\partial x}$$

### A.2   Gradient $\odot$ Input

Gradient element-wise product with the input. Ancona et. al. show that this input gradient product is equivalent to DeepLift, and $\epsilon$-LRP (other explanations methods), for a network with with only Relu(s) and no additive biases.

$$E_{\text{grad}\odot\text{input}}(x) = x \odot \frac{\partial S}{\partial x}$$

### A.3   Guided Backpropagation (GBP)

GBP specifies a change in how to back-propagate gradienst for ReLus. Let $\{f^l, f^{l-1}, ..., f^0\}$ be the feature maps derived during the forward pass through a DNN, and $\{R^l, R^{l-1}, ..., R^0\}$ be 'intermediate representations' obtained during the backward pass. Concretely, $f^l = relu(f^{l-1}) = max(f^{l-1}, 0)$, and $R^{l+1} = \frac{\partial f^{out}}{\partial f^{l+1}}$ (for regular back-propagation). GBP aims to zero out negative gradients during computation of $R$. The mask is computed as:

$$R^l = 1_{R^{l+1}>0} 1_{f^l>0} R^{l+1}$$

$1_{R^{l+1}>0}$ means keep only the positive gradients, and $1_{f^l>0}$ means keep only the positive activations.

### A.4   GradCAM and Guided GradCAM

Introduced by Selvaraju et al. [19], GradCAM explanations correspond to the gradient of the class score (logit) with respect to the feature map of the last convolutional unit of a DNN. For pixel level granularity GradCAM, can be combined with Guided Backpropagation through an element-wise product.

Following the exact notation by Selvaraju et al. [19], let $A^k$ be the feature maps derived from the last convolutional layer of a DNN. Consequently, GradCAM is defined as follows: first, neuron importance weights are calculated, $\alpha_c^k = \frac{1}{Z} \sum_i \sum_j \frac{\partial S}{\partial A_{ij}^k}$, then the GradCAM mask corresponds to: $ReLU(\sum_k \alpha_c^k A^k)$. This corresponds to a global average pooling of the gradients followed by weighted linear combination to which a ReLU is applied. Now, the Guided GradCAM mask is then defined as:

$$E_{\text{guided}-\text{gradcam}}(x) = E_{\text{gradcam}} \odot E_{\text{gbp}}$$

### A.5   Integrated Gradients (IG)

IG is defined as:

$$E_{\text{IG}}(x) = (x - \bar{x}) \times \int_0^1 \frac{\partial S(\bar{x} + \alpha(x - \bar{x})}{\partial x} d\alpha$$

where $\bar{x}$ is the baseline input that represents the absence of a feature in the original sample $x_t$. $\bar{x}$ is typically set to zero.

## A.6  SmoothGrad

Given an explanation, $E$, from one of the methods previously discussed, a sample $x$, the SmoothGrad explanation, $E_{\mathrm{sg}}$, is defined as follows:

$$E_{\mathrm{sg}}(x) = \frac{1}{N} \sum_{i=1}^{N} E(x + g_i),$$

where noise vectors $g_i \sim \mathcal{N}(0, \sigma^2))$ are drawn i.i.d. from a normal distribution.

## A.7  VarGrad

Similar to SmoothGrad, and as referenced in [37] a variance analog of SmoothGrad can be defined as follows:

$$E_{\mathrm{vg}}(x) = \mathcal{V}(E(x + g_i)),$$

where noise vectors $g_i \sim \mathcal{N}(0, \sigma^2))$ are drawn i.i.d. from a normal distribution, and $\mathcal{V}$ corresponds to the variance. In the visualizations presented here, explanations with VG correspond to the VarGrad equivalent of such masks. Seo et al. [38] theoretically analyze VarGrad showing that it is independent of the gradient, and captures higher order partial derivatives.

# B  DNN Architecture, Training, Randomization & Metrics

**Experimental Details Data sets & Models.** We perform our randomization tests on a variety of datasets and models as follows: an Inception v3 model [39] trained on the ImageNet classification dataset [40] for object recognition, a Convolutional Neural Network (CNN) trained on MNIST [41] and Fashion MNIST [42], and a multi-layer perceptron (MLP), also trained on MNIST and Fashion MNIST.

**Randomization Tests** We perform 2 types of randomizations. For the model parameter randomization tests, we re-initialized the parameters of each of the models with a truncated normal distribution. We replicated these randomization for a uniform distribution and obtain identical results. For the random labels test, we randomize, completely, the training labels for a each-model dataset pair (MNIST and Fashion MNIST) and then train the model to greater than 95 percent training set accuracy. As expected the performance of these models on the tests set is random.

**Inception v3 trained on ImageNet.** For Inception v3, we used a pre-trained network that is widely distributed with the tensorflow package available at: `https://github.com/tensorflow/models/tree/master/research/slim#Pretrained`. This model has a 93.9 top-5 accuracy on the ImageNet test set. For the randomization tests, we re-initialized on a per-block basis. As noted in [43], each inception block consists of multiple filters of different sizes. In this case, we randomize all the the filter weights, biases, and batch-norm variables for each inception block. In total, this randomization occurs in 17 phases.

**CNN on MNIST and Fashion MNIST.** The CNN architecture is as follows: input -> conv (5x5, 32) -> pooling (2x2)-> conv (5x5, 64) -> pooling (2x2) -> fully connected (1024 units) -> softmax (10 units). We train the model with the ADAM optimizer for 20 thousand iterations. All non-linearities used are ReLU. We also apply weight decay (penalty 0.001) to the weights of the network. The final test set accuracy of this model is 99.2 percent. For model parameter randomization test, we reinitialize each layer successively or independently depending on the randomization experiment. The weight initialiazation scheme followed was a truncated normal distribution (mean: 0, std: 0.01). We also tried a uniform distribution as well, and found that our results still hold.

**MLP trained on MNIST.** The MLP architecture is as follows: input -> fully connected (2500 units) -> fully connected (1500 units) -> fully connected (500 units) -> fully connected (10 units). We also train this model with the ADAM optimizer for 20 thousand iterations. All non-linearities used are Relu. The final test set accuracy of this model is 98.7 percent. For randomization tests, we reinitialize each layer successively or independently depending on the randomization experiment.

**Inception v4 trained on Skeletal Radiograms.** We also analyzed an inception v4 model trained on skeletal radiograms obtained as part of the pediatric bone age challenge conducted by the radiological society of north America. This inception v4 model was trained retained the standard original parameters except it was trained with a mixed L1 and L2 loss. In our randomization test as indicated in figure 1, we reinitialize all weights, biases, and variables of the model.

**Calibration for Similarity Metrics.** As noted in the methods section, we measure the similarity of the saliency masks obtained using the following metrics: Spearman rank correlation with absolute value (absolute value), Spearman rank correlation without absolute value (diverging), the structural similarity index (SSIM), and the Pearson correlation of the histogram of gradients (HOGs) derived from two maps. The SSIM and HOGs metrics

are computed for ImageNet explanation masks. We do this because these metrics are suited to natural images, and to avoid the somewhat artificial structure of Fashion MNIST and MNIST images. We conduct two kinds of calibration exercises. First we measure, for each metric, the similarity between an explanation mask and a randomly sampled (Uniform or Gaussian) mask. Second, we measure, for each metric, the similarity between two randomly sampled explanation masks (Uniform or Gaussian). Together, these two tasks allow us to see if high values for a particular metric indeed correspond to meaningfully high values.

We use the skimage HOG function with a (16, 16) pixels per cell. Note that the input to the HOG function is 299 by 229 with the values normalized to [-1, +1]. We also used the skimage SSIM function with a window size of 5. We obtained the gradient saliency maps for 50 images in the ImageNet validation set. We then compare these under the two settings described above; we report the average across these 50 images as the following tuple: (Rank correlation with no absolute value, Rank correlation with absolute value, HOGs Metric, SSIM). The average similarity between the gradient mask and random Gaussian mask is: $(-0.00049, 0.00032, -0.0016, 0.00027)$. We repeat this experiment for Integrated gradient and gradient$\odot$input and obtained: $(0.00084, 0.00059, 0.0048, 0.00018)$, and $(0.00081, 0.00099, -0.0024, 0.00023)$. We now report results for the above metrics for similarity between two random masks. For uniform distribution [-1, 1], we obtain the following similarity: $(0.00016, -0.0015, 0.078, 0.00076)$. For Gaussian masks with mean zero and unit variance that has been normalized to lie in the range [-1, 1], we obtain the following similarity metric: $(0.00018, 0.00043, -0.0013, 0.00023)$.

## C   Additional Figures

We now present additional figures referenced in the main text.

Figure 7: **No observable difference in explanations B & C.** A): a skeletal radiogram from the pediatric bone age challenge organized by the radiological society of north America (RSNA). Given several thousand radiographs, challenge participants are tasked with building models to predict the age (in months) of the patient. B) Guided Backprop explanation of sample A for an Inception v4 model trained on the radiograms. C) Guided Backprop explanation of sample A for an Inception v4 model **with completely random weights**. Both explanations are virtually indistinguishable.

Figure 8: **Large Version: Figure showing saliency output along with edge detector for each image. We also include the SSIM and HOGs similarity metric.**

Figure 9: Model parameter randomization test

Figure 10: **Input ⊙ Random gradient experiment.**

Figure 11: Rank Correlation Metric (with absolute values) on all architectures and datasets.

## Rank Correlation - No Absolute Values

Figure 12: **Rank Correlation Metric (without absolute values) on all architectures and datasets.**

Figure 13: **Comparison between explanations for a true model and one trained on random labels. MLP on MNIST**

**CNN - FMNIST**

**Absolute-Value Visualization**     **Diverging Visualization**

Figure 14: **Comparison between explanations for a true model and one trained on random labels. CNN on Fashion MNIST.**

**MLP - FMNIST**

**Absolute-Value Visualization**     **Diverging Visualization**

Figure 15: **Comparison between explanations for a true model and one trained on random labels. MLP on Fashion MNIST**

Figure 16: Cascading Randomization for Corn Class Image.

Figure 17: Cascading Randomization for Corn Class Image: Diverging Visualization.

Figure 18: Cascading Randomization for Dog Image.

Figure 19: Cascading Randomization for Dog Image: Diverging Visualization.

Figure 20: Independent Randomization for Corn.

Figure 21: Independent Randomization for Corn.

**CNN MNIST**

Figure 22: **Independent and successive re-initialization for CNN trained on MNIST. Left**: independent randomization of each layer of the CNN. **Right**:successive randomization of each layer of the CNN. **Note**: VG represents Vargrad (See. Methods section in appendix for method definition.)

**CNN MNIST**

Figure 23: **Independent and successive re-initialization for CNN trained on MNIST. Left**: independent randomization of each layer of the CNN. **Right**:successive randomization of each layer of the CNN.

Figure 24: **Independent and successive re-initialization for MLP (3-hidden layers) trained on MNIST. Left**: independent randomization of each layer of the MLP. **Right**: successive randomization of each layer of the MLP.

Figure 25: **Independent and successive re-initialization for MLP (3-hidden layers) trained on MNIST. Left**: independent randomization of each layer of the MLP. **Right**: successive randomization of each layer of the MLP.

Figure 26: Saliency Methods on a 1-layer Convolutional Sum Model.

Figure 27: **Top**: Saliency maps for an input image. **Bottom**: Saliency maps for an input ⊙ edge detector. This corresponds to an input where all the non-edges have been zeroed out. We see that, qualitatively, both maps look visually similar for GBP and Guided GradCAM.

Figure 28: **A**: Reverse Cascading Randomization. **B**: Cascading Randomization on MNIST for DeepLIFT (Orange) and $\epsilon$-LRP (Green). **C**: Cascading Randomization (no absolute value for spearman metric) on MNIST for DeepLIFT (Orange) and $\epsilon$-LRP (Green). **D**: Cascading randomization for the perturbation method ([20]. **E**: Spearman Rank Correlation metric for perturbation method ([20].