[Reviews · NeurIPS 2018]

Reviewer 1



Summary --- This paper introduces a new set of criteria for evaluating saliency/heatmap visual explanations of neural net predictions. It evaluates some existing explanation approaches and shows that some of them do not pass these simple tests. The first test compares explanations of a base model to explanations of the same model with its weights randomly re-initialized. If the same visualization results from both sets of weights then it does not reflect anything the neural net has learned. Similarly, the second test compares a base model to the same model trained with randomly permuted labels. In this case the weights are not randomly initialized, but still cannot contain much information specific to the task (image classification) because the labels were meaningless. For both tests pairs of visualizations are compared by manual inspection and by computing rank correlation between the two visualizations. Layers are re-initialized starting from the output and moving toward the input, generating an explanation for each additional layer. Techniques including Integrated Gradients, Guided BackProp, and Guided GradCAM produce very similar explanations whether all parameters are random or not. Other techniques (Gradient and GradCAM) drastically change the explanations produced after the first layer is randomized and continue to produce these different and appropriately noisy explanations as more layers are randomized. The surprising conclusion this work offers is that some methods which attempt to explain neural nets do not significantly change their explanations for some significantly different neural nets. To explain this phenomena it analyzes explanations of a linear model and of a single conv layer, claiming that the conv layer is similar to an edge detector. It also acknowledges that architectural priors play a significant role in the performance of conv nets and that random features are not trivial. Strengths --- I buy the basic idea behind the approach. I think this is a good principle to use for evaluating saliency style visual explanations, although it doesn't act as an ultimate form of quality, just potentially useful initial sanity checks. The criteria has clarity that other approaches to saliency evaluation lack. Overall, the set of experiments is fairly large. Weaknesses --- None of these weaknesses stand out as major and they are not ordered by importance. * Role of and relation to human judgement: Visual explanations are useless if humans do not interpret them correctly (see framework in [1]). This point is largely ignored by other saliency papers, but I would like to see it addressed (at least in brief) more often. What conclusions are humans supposed to make using these explanations? How can we be confident that users will draw correct conclusions and not incorrect ones? Do the proposed sanity checks help identify explanation methods which are more human friendly? Even if the answer to the last question is no, it would be useful to discuss. * Role of architectures: Section 5.3 addresses the concern that architectural priors could lead to meaningful explanations. I suggest toning down some of the bolder claims in the rest of the paper to allude to this section (e.g. "properties of the model" -> "model parameters"; l103). Hint at the nature of the independence when it is first introduced. Incomplete or incorrect claims: * l84: The explanation of GBP seems incorrect. Gradients are set to 0, not activations. Was the implementation correct? * l86-87: GradCAM uses the gradient of classification output w.r.t. feature map, not gradient of feature map w.r.t. input. Furthermore, the Guided GradCAM maps in figure 1 and throughout the paper appear incorrect. They look exactly (pixel for pixel) equivalent to the GBP maps directly to their left. This should not be the case (e.g., in the first column of figure 2 the GradCAM map assigns 0 weight to the top left corner, but somehow that corner is still non-0 for Guided GradCAM). The GradCAM maps look like they're correct. l194-196: These methods are only equivalent gradient * input in the case of piecewise linear activations. l125: Which rank correlation is used? Theoretical analysis and similarity to edge detector: * l33-34: The explanations are only somewhat similar to an edge detector, and differences could reflect model differences. Even if the same, they might result from a model which is more complex than an edge detector. This presentation should be a bit more careful. * The analysis of a conv layer is rather hand wavy. It is not clear to me that edges should appear in the produced saliency mask as claimed at l241. The evidence in figure 6 helps, but it is not completely convincing and the visualizations do not (strictly speaking) immitate an edge detector (e.g., look at the vegitation in front of the lighthouse). It would be useful to include a conv layer initialized with sobel filter and a canny edge detector in figure 6. Also, quantitative experimental results comparing an edge detector to the other visual explanations would help. Figure 14 makes me doubt this analysis more because many non-edge parts of the bird are included in the explanations. Although this work already provides a fairly large set of experiments there are some highly relevant experiments which weren't considered: * How much does this result rely on the particular (re)intialization method? Which initialization method was used? If it was different than the one used to train the model then what justifies the choice? * How do these explanations change with hyperparameters like choice of activation function (e.g., for non piecewise linear choices). How do LRP/DeepLIFT (for non piecewise linear activations) perform? * What if the layers are randomized in the other direction (from input to output)? Is it still the classifier layer that matters most? * The difference between gradient * input in Fig3C/Fig2 and Fig3A/E is striking. Point that out. * A figure and/or quantitative results for section 3.2 would be helpful. Just how similar are the results? Quality --- There are a lot of weaknesses above and some of them apply to the scientific quality of the work but I do not think any of them fundamentally undercut the main result. Clarity --- The paper was clear enough, though I point out some minor problems below. Minor presentation details: * l17: Incomplete citation: "[cite several saliency methods]" * l122/126: At first it says only the weights of a specific layer are randomized, next it says that weights from input to specific layer are randomized, and finally (from the figures and their captions) it says reinitialization occurs between logits and the indicated layer. * Are GBP and IG hiding under the input * gradient curve in Fig3A/E? * The presentation would be better if it presented the proposed approach as one metric (e.g., with a name), something other papers could cite and optimize for. * GradCAM is removed from some figures in the supplement and Gradient-VG is added without explanation. Originality --- A number of papers evaluate visual explanations but none have used this approach to my knowledge. Significance --- This paper could lead to better visual explanations. It's a good metric, but it only provides sanity checks and can't identify really good explanations, only bad ones. Optimizing for this metric would not get the community a lot farther than it is today, though it would probably help. In summary, this paper is a 7 because of novelty and potential impact. I wouldn't argue too strongly against rejection because of the experimental and presentation flaws pointed out above. If those were fixed I would argue strongly against rejection. [1]: Doshi-Velez, Finale and Been Kim. “A Roadmap for a Rigorous Science of Interpretability.” CoRR abs/1702.08608 (2017): n. pag.

Reviewer 2



The paper aims to answer to often-asked questions for assessing the quality of an explanation method for complex ML algorithms. This is in-line with an increasing amount of literature on assessment of explanation methods such as what explanation method should achieve, such as • Lundberg, Scott M., and Su-In Lee. "A unified approach to interpreting model predictions." Advances in Neural Information Processing Systems. 2017. • Kindermans, Pieter-Jan, et al. " Learning how to explain neural networks: PatternNet and PatternAttribution." ICLR. 2018. • Their main contribution are two simple strategies for testing the faith of the explanation method to the model and the data. For testing model faithness, they propose comparing the explanation output of a trained model to that of a randomly initialized one. For faith to the data, labels are randomly shuffled for training. They come to the conclusion, that many of the currently used methods are independt of data and model to be explained. Additionally they compare outputs of common explainability methods to that of a Sobel edge detector. The methods they propose are easy to implement and provide a valid and intuitive ‘sanity check’ for two conditions that all explainability methods should fulfill. There are extensive experiments, applying these two methods to a variety of often used explainability methods. The writing style is very clear and the paper has a clear logical structure apart from the problems mentioned. The authors motivate and explain the problem, they are solving. A weakness of the paper is, that the analysis is often restricted to qualitative instead of quantitative statements. Starting with l. 246 they argue that edge detector images look very similar to the output of many explanation methods. This in itself is not surprising, as NN do process edges and shapes as well. Simple correlation analysis of edges and explanation output would give more insight into whether NN do not prioritize important edges in the input. The structure of the paper is lacking. The discussion section includes more experiments and does not actually interpret or discuss the earlier results. In my opinion this paper proposes two sound methods for assessing explainability methods. Unfortunately there is no significant discussion or proposed explanation of the results they found. Due to this and the fact that very easily obtained quantitative measurements are missing, this paper is marginally below acceptance Minor: l. 101 existing, not exisiting 190 sentence incomplete “Learning to explain …” is published at ICML – replace arxive citation

Reviewer 3



Content: In the field of deep learning a large number of "saliency methods" has been proposed over the last years, that is, methods that try to explain which parts of an input image drive the network's decision for some label. The authors of the present submission propose a few sanity checks that saliency methods should pass as a necessary condition for them to explain anything about the model decisions. The propose two classes of tests: Model parameter randomization tests and data randomization tests. The first class asserts that replacing all or some model parameters with randomly initialized parameters (and therefore degrading model performance) should change the saliency method's output. The second class asserts that randomly permuting labels in the training data (and therefore forcing the model to memorize the training samples) should change the saliency output. They apply these tests to a variety of commonly used saliency methods and show that many methods fail in some or all of the tests. Finally they offer potential explanations for some of the failures with the example of simple linear or one-layer convolutional models. Quality: In general I think the presented submission is of high quality. The proposed tests are mainly well-motived (but see below), the experiments seem reproducible. With respect to the data randomization test, the authors claim that an insensitivity to the permuted labels shows that the method doesnot depend on the relationship between instances and labels in the original data. I'm not completely convinced by this argument: Randomization of the labels forces the network to memorize the input images together with their labels. It could be the case that the features that help classifying an image are the same that help memorizing the image. But I agree with the authors that one would hope to see a difference. In the discussion, the authors first focus on methods that approximate an element-wise product of input and gradient and argue that in these cases the input "structure" dominates the gradient (line 179). This is shown for MNIST in Figure 5. In general I agree with the authors, but I have some issues with this statement and the presented evidence for it: * The shown effect mainly holds for MNIST data where large areas of the image are black. In my opinion the actual issue here is that all versions of "input times gradient" assign a special role to the value zero in the input. In MNIST that even makes some sense (since here zero is the absense of a stroke), but for natural images in general I'm not convinced by given zero (i.e. black) a special role compared to gray or white. For me this is the main reason to avoid input-times-gradient methods and the observed effect is just a side-effect of that. * On a related note, in Figure 5d: Is scale of the noise on the x-axis relative to the scale of the input data? If so, at least for imagenet (which is, unlike MNIST, not sparse) I would expect that for sufficiently large gradient noise the noise significantly changes the rank correlation. In section 5.2 the authors analyze two simple models: a linear model and a one-layer convolutional network. They show that the convolutional model mainly produces saliency maps that look like the output of edge detectors and argue that this is a problem for the saliency methods. I don't agree completely with this: a simple one-layer convolutional network in the end is actually a combination of local pooling and edge detection. So seeing edges in the model explanation might not be a failure of the explanation method after all Clarity: The submission is presented very clearly and structured very well. I enjoyed reading it. I have a few minor concerns: * Figure 2 is never referenced in the main text and I'm not sure which model is used in this figure. * I first wondered how the models in the reparametrization test are (re)-initialized, then I found the detail in the supplement. A short reference to the supplement might be helpful if the space allows. Originality: The proposed sanity checks are original. Randomizing model weights or training labels is something that has been done often before, but utilizing these methods as sanity checks for saliency methods is a new contribution for the field of saliency methods. The submission mentions other related methods like masking salient areas of the image to change the model decisions (but see below). Significance: I like the proposed sanity checks and think they can be very helpful as a necessary condition for estimating the explaining power of existing saliency methods. That makes this work a relevant contribution. However, I'm missing a few things that I think would increase the relevance and significance of this submission. My main concern is that all included saliency methods are different versions of gradients or gradients times input, which are known to have weaknesses (See e.g. Kindermans et al, Learning how to explain neural networks: PatternNet and PatternAttribution, ICLR 2018). Therefore I would like to see methods that explicitly try to overcome these limitations. This includes that aforementioned PatternNet which is nicely theoretically founded and also Zintgraf et al, Visualizing Deep Neural Network Decisions: Prediction Difference Analysis, ICLR 2017. Another easy-to-fix issue that in my opinion degrades the high potential impact of this submission: Right now there is no summary which of the included saliency methods the authors would recommend for future use given their results. I think this would be an important conclusion for the paper If the two above points are fixed, I would argue that the presentd work is of very high relevance to the community and definitely should be published. In the current state the work is still relevant because the proposed sanity checks are very well motivated but the significance is somewhat limited. One additional point that is less important in my opinion, but would help underline the relevance of the work: the submission mentions other related methods like masking salient areas of the image to change the model decisions (l. 107). I think it would be interesting to see whether the results of the submission are consistent with these other quality measures. This would show whether the proposed sanity checks offers new evidence over those methods that should be more than just sanity checks. Minor: l 17: "Cite several saliency methods": I guess that's a leftover todo l 254" I guess the reference to Figure 14 is actually supposed to refer to Figure 11? Update after the rebuttal: I'm pretty happy with the authors response. I like especially that they now include more methods and some results on ImageNet. I'm still missing a thorough discussion of the label permutation test (and would love to see more ImageNet results due to the special structure of MNIST). Nevertheless I think this paper has an important contribution to make and therefore I'm upgrading my rating to an 8.